# ‘Care Beyond Co-Residence’: A Qualitative Exploration of Emotional and Instrumental Care Gaps Among Older Adults in Migrant Households of Kerala

**DOI:** 10.3390/ijerph22111745

**Published:** 2025-11-18

**Authors:** Anu Mohan, Teddy Andrews Jaihind Jothikaran, Divya Sussana Patil, Lena Ashok

**Affiliations:** 1Department of Social and Health Innovation, Prasanna School of Public Health, Manipal Academy of Higher Education, Manipal 576104, India; anu.mohan@learner.manipal.edu (A.M.); lena.ashok@manipal.edu (L.A.); 2Department of Health Technology and Informatics, Centre for Evidence-Informed Decision-Making, Prasanna School of Public Health, Manipal Academy of Higher Education, Manipal 576104, India; divya.patil@manipal.edu

**Keywords:** aging, left-behind older adults, distance care, emotional deprivation, functional deprivation, vulnerability

## Abstract

The convergence of youth migration and the nuclearization of families has altered conventional living arrangements in India, indicating a sharp rise in the number of families in which older adults live alone due to the outmigration of their adult children. This study aims to explore the perceptions of left-behind older adults regarding long-distance care practices by their adult children and to describe the practical and functional care deficits that lead to vulnerability and unmet mental health care in migrant households. Twenty older adults above 65 years of age living alone or with a spouse for at least one year due to the out-migration of their adult children were selected purposively. The analysis revealed that distance from migrant children makes older adults feel anxious, miss their family togetherness, and experience increased loneliness and care gaps in later years, contributing to a multifaceted causality of vulnerability while aging alone. Narratives of distance care are often shaped by the bidirectional flow of care across generations through virtual and in-person means, where emotional and functional deprivations continue to challenge the quality of informal distant care among left-behind older adults. Mental health promotion among community-dwelling older adults is crucial for sustaining their functional capacity, thereby delaying psychological morbidities during aging.

## 1. Background

India has witnessed a significant increase in longevity over the past several decades, from approximately 32 years in 1951 to over 70 years in 2025, indicating opportunities and challenges in social, economic, and health planning [1]. The postpandemic era witnessed more than a 70% increase in youth outmigration owing to various push and pull factors influencing both internal and international migration [2]. The convergence of youth migration and the nuclearization of families has altered conventional living arrangements in India, indicating a sharp rise in the number of families where older adults live alone or with their spouses due to the outmigration of their adult children. Findings from the Longitudinal Aging Survey in India (2018) and Building Knowledge Base in Population Aging in India (2011) estimate that between 36% and 43% of older adults are empty nesters and are not coresiding with any of their children [3,4]. Kerala has the highest proportion of elderly individuals in India, with an estimated proportion of up to 20% by 2026, mirroring trends in developed countries, such as a rising dependency ratio, emergence of neglect, loneliness, and increased demand for assisted living [5].

Although the migration of adult children has improved the living conditions of older parents through financial remittance, access to advanced communication devices, and provisions for paid care, existing evidence reveals functional deprivations and emotional deficits in long-distance care, which is further compounded by declining physical and cognitive capabilities in later years [6,7,8]. According to Scheffel & Zhang, (2019), the migration of adult children has reduced happiness among older parents by 6.6%, leading to an increase in loneliness of 3.3% [9]. Furthermore, living alone in later years has impacted activities of daily living (ADL) and social connections among older adults, increasing the risk of loneliness among those age alone [10]. Ugargol & Bailey (2020) confirmed a significant association between living arrangements and functional status, health care patterns, disability and physical decline among left-behind older adults in South India [11]. Empirical studies on living arrangements and well-being in later years argue that older adults in migrant households experience added care deprivation due to limited in-person care, risks of social isolation, and neglect, indicating a greater risk of vulnerability across physical, emotional, and economic dimensions of health and well-being than their counterparts residing with their families do [12,13,14,15]. Furthermore, older adults with a migrant son are found to have a 26% significantly greater likelihood of being depressed than older adults with a non-migrant son, indicating an increase in mental health concerns among community-dwelling older adults who age outside intergenerational coresidence [13]. Thapa et al. (2018) further confirmed that older parents who age alone have higher levels of loneliness and depressive symptoms, lower life satisfaction, weaker cognitive ability, and poorer psychological health than their counterparts who reside with their kin. While recent evidence points to loneliness and social isolation as key symptoms of depression, very limited exploration has been conducted to conceptualize the public health crisis of social disconnection from a South Asian perspective [14], where experiences of loneliness contribute to vulnerability while aging alone [16]. A decline in mental health has proven to pose serious public health risks among older adults, emphasizing positive correlations with premature mortality, cognitive decline and cardiovascular diseases on par with classic medical risk factors and warranting further investigation into the context and environmental support of aging alone [17]. Age alone is widely recognized as a global public health risk, as it implies delayed treatment seeking, increased odds of morbidity and mortality, and increased risks of frailty and disability, which in turn increases the burden on long-term care and rehabilitation [18].

Theories on transnationalism have attempted to explore how families and individuals maintain multiple economic, cultural, and social ties that transcend national borders and continue to influence the circulation of care that crosses borders [19,20]. Furthermore, theories on the circulation of care are often described as the transfer of caregiving resources, including emotional support, financial resources, and practical assistance in transnational families [21]. These circulations are often mediated through communication, remittances, visits, and the arrangement of alternative care responsibilities, enabling families to sustain care practices and relationships despite physical distance. Thus, virtual care is defined as the act of using technology to connect a care provider and a receiver, especially when they are separated due to distance or mobility restrictions [22]. Further application of the social exchange perspective underscores the role of reciprocal reward-cost dynamics in fostering relationships, even from a distance [7,14]. Existing evidence argues that migration may hinder the intergenerational reciprocity of care and filial piety, posing a high risk of discontent in the relationship between migrant children and their left-behind parents in the homeland [23]. From a lifespan perspective, the social relationship exchange framework perceives loneliness as a mismatch between the expected and actual relationships in later years, outlining six social relationship expectations: intimacy, proximity, support, fun, generativity, and respect that, if unmet, can contribute to loneliness later in life [24]. However, existing conceptualizations conclude an evident gap in theory-driven research on experiencing loneliness among older adults, warranting more conclusive research into the experiences of vulnerability while aging alone. Emotional care gaps are deprivations in fulfilling unmet care needs, which are often intensified by loneliness [25], whereas instrumental care gaps increase with unmet needs in instrumental activities of daily living (IADLs) and mobility [26]. Given that care is a dynamic and negotiated concept across cultures, limited theories could fully capture the nuanced realities of long-distance care. The degree of care varies depending on the unique experience and subjective perceptions, calling for an in-depth analysis of the experiences of “care”that extends beyond coresidence in migrant housheolds and how their care needs can be integrated into care delivery models [27]. Previous research has extensively attempted to measure and correlate psychological distress across various living arrangements, indicating a high risk of vulnerability among older adults in solitary living. However, many investigations have not yet been conducted to describe the specific emotional and functional effects of youth migration on older parents, including their practical deficits, loneliness, and anxiety, which are context-specific and culturally responsive [24,28,29]. Furthermore, existing research often overlooks the unique needs and perceptions of aging alone among older adults across different age groups, comorbid statuses, and health conditions [30,31,32]. Despite the alarming rise in the proportion of older adults in Kerala, existing research is often restricted to sample surveys, which quantify the prevalence of various psychosocial dimensions that foster positive aging, leaving room for an inductive approach in understanding the sociocultural nuances of vulnerability and challenges in meeting the mental health care needs of older adults who are left behind in migrant households. Moreover, the South Asian landscape of care is deeply rooted in mutual care obligations and the normative exceptions of filial piety, offering a unique conceptualization of care that crosses borders. Thus, the present study aims to explore how adults in migrant households perceive the long-distance care practices of their adult children and to narrate the practical and functional care deficits that lead to vulnerability in remote care.

## 2. Materials and Methods

### 2.1. Study Design

A qualitative narrative approach was employed to explore and analyze the experiences of long-distance care among left-behind older adults. This paper is part of a larger study on the perceptions and experiences of left-behind older adults, their caregivers, and service providers in long-distance care. However, the present study uses a subset of data on participants’ experiences of living alone while managing their lives without immediate support and care from their children. A narrative design includes stories shared by participants, aiming to explore how participants construct meaning from their lived experiences [32]. A narrative inquiry is used to understand how emotional and instrumental support is perceived and experienced by older adults in migrant households of Kerala.

### 2.2. Study Setting

The study was carried out in three geographic zones of Kerala, including Kannur [North Kerala], Thrissur [Central Kerala], and Pathanamthitta [South Kerala], based on the population of emigrants, the number of older adults and the proportion of elderly individuals living alone. Pathanamthitta is widely known as a remittance hub due to high rates of international migration, while Thrissur is considered to have a balance of internal and international migration, with the highest number of student migrants, and Kannur is traditionally an early center for Gulf migration; however, it is now witnessing a decline in emigration but continues to be the emigration hub in northern Kerala [31,32,33]. The Kumbanad region of Koipuram Grama Panchayath, Arimpur Panchayath, and Koothuparambu municipality was selected from each district purposively based on the geographical location and demographic features.

### 2.3. Participants and Sampling

Twenty older adults above 65 years of age living alone or with a spouse for at least one year due to the out-migration of their adult children were selected purposively. Older adults with serious physical and mental impairments, older adults in intergenerational co-residence, and those with children who have migrated within Kerala were excluded from the study; thus, interstate and overseas migration of children was included. The participants were contacted primarily through community gatekeepers, including representatives of local self-government organisations, officials of associations for senior citizens, and local leaders from each religious community, and through the snowball effect initiated by participants to ensure heterogeneous representation of participants.

The characteristics of the participants are summarized in Table 1.

### 2.4. Data Collection

A semistructured in-depth interview guide was prepared and then translated into Malayalam [vernacular language]. Data collection was performed from April 2023 to August 2023. Each interview took approximately 45–50 min. The protocol and tool were approved by the Institutional Ethics Committee of Kasturba Medical College and Kasturba Hospital IEC1:248/2022. Eight female and twelve male participants between the ages of 65 and 83 were interviewed until data saturation. The interviews were recorded after written informed consent was obtained from the participants in accordance with the Declaration of Helsinki. The participants were asked to share their sociodemographic details [age, annual income, living arrangement, occupation, health status, and frequency of visits by children], perceptions, and experiences living alone and receiving care from a distance. Questions such as *“Can you describe the experiences of living apart from your children?” “How do they care for you from afar?”* and *“What are the major differences in lifestyle before and after their migration?”* were asked to obtain older adults’ insights into living alone and receiving virtual care. Field notes were prepared to capture additional information along with the interview recordings.

### 2.5. Data Analysis

The audio recordings of the interviews were transcribed and translated into English manually. Thematic analysis was employed to analyse the interview data, following the six-step framework outlined by Braun and Clarke [34] These steps included: (1) familiarization with the data, (2) generating initial codes, (3) searching for themes, (4) reviewing themes, (5) defining and naming themes, and (6) producing the final report. The organization and management of codes and themes were performed via NVivo version 29. A thematic analysis was performed to derive deeper insights into the experience of long-distance care later in life among left-behind older adults in Kerala [34,35,36]. The analysis revealed five major themes that shape the essence of solitary living and long-distance care received in migrant households: (1) cues of care from afar, (2) missing coexistence with multigenerational families, (3) feeling lonely, neglected and stagnant, (4) feeling anxious while aging alone, and (5) deficits in instrumental and health needs.

## 3. Findings

The major themes and codes used in the analysis are summarized in Table 2.

The conceptual model derived from the participants’ narratives, capturing the interplay of virtual care, emotional deprivation, and instrumental care gaps among left-behind older adults, is illustrated in Figure 1. Left-behind older adults continue to expect practical and emotional support from their adult children and extended families. Although migrant children attempt to fulfill their care virtually and through occasional in-person visits, the absence of coresidence elevates feelings of loneliness, leading to anxiety in aging alone. On the other hand, practical deficits in managing daily tasks and accessing healthcare further intensify the feeling of anxiety, suggesting the interplay of emotional and instrumental deficits in shaping the experience of vulnerability in remote care. Furthermore, minimal support from the microsystem exacerbates the sense of isolation, whereas deficits in instrumental and health care needs may diminish perceived filial fulfillment, highlighting the complex interconnections of vulnerability that persist among older adults left behind due to intergenerational distance.

### 3.1. Cues of Care from Afar

The participants’ perceptions of care seem to have been ambiguous, depending on how they experienced care cues from either migrant children or acquaintances nearby. For several participants, virtual closeness makes them feel cared for despite distance. Regular phone calls, video conferencing, and efforts to stay updated about each other’s lives were regarded as crucial cues of care when distance was inevitable. One of the participants stated the following:


*“My son always calls me before leaving for duty. Because his time is different, it is early morning there. My daughter lives in a time zone five and a half hours behind, so she calls in the evening. She always makes sure to call before I go to bed. She cannot pass a day without enquiring about me”*
(OA-2)

In addition to maintaining regular contact, arranging alternative care provisions in their absence seems to make several participants feel cared for from afar. Reminding them of an approaching consultation, speaking to the doctor on their behalf, arranging transport to hospitals, and virtual monitoring are some reasons for participants to feel cared for and loved, even when their children cannot be there in person. An example was provided by one of the participants.


*“Their call pattern itself makes me feel they are bothered. They do video calls now and then. Keeps on asking me “if everything is all right”. Even at our slightest discomfort, they tend to be more anxious than me. When it is time for a hospital consultation, they keep a check on dates, arrange vehicles if they are not around, speak to a doctor, and keep checking their health status virtually. Even if they are apart, they try to do whatever they can do, and hence, I feel distance is not a very big crisis for us.”*
(OA-7)

Although several participants considered arranging alternative care provisions to be influential in defining care, many others considered children coming down during times of need as an expression of care in trans local families. In this context, participants whose children come down to care for them during hospitalizations stated as follows.


*“I had to undergo surgery due to goiter, and for that, my son extended his leave. Also Additionally, my daughter-in-law stayed back to look after me, as I could not carry out things by myself. Therefore, I know they always care. My children keep on asking, “How do you feel now?” My daughter is very much concerned about my health status and keeps checking on me even though she is busy with her research and labs.”*
(OA-8)

Thus, older adults who age alone perceive the efforts of children to compensate for physical distance as cues of care from afar. Several participants rationalise distance by staying connected with their kin through virtual transactions, adapting to alternative care arrangements, and being hopeful that children will come down when needed.

### 3.2. Missing the Coexistence of Multigenerational Families

Although virtual care has diminished the distance between households, physical absence continues to affect care transactions, often leaving a void that may impact both caregivers and recipients. Amidst all the technological advancements, several participants long for the sense of togetherness, in-person care, and the joy of watching their children and grandchildren grow up before their eyes. Although most of the participants rationalized their children’s decision to migrate, many of them experienced a sense of missing intergenerational coresidence. One of the participants stated the following.


*“When they are around, we feel more alive. Like all others, we also wish to live by their side, sharing love with the grandchildren. However, I must wait years to see my grandchildren who are abroad. I missed watching them growing up, learning, and playing. I know not having them is always a loss that can never be replaced.”*
(OA-1)

Several participants are longing not only for their children and grandchildren but also for the happiness and fulfillment they experienced before migration, when everyone lived together. According to a male participant, migration has diminished the quality of family time, as stated below.


*“It is been a very long time since we sat together, discussing family matters, or even playing cards. This used to happen regularly during their childhood when they were around. Now each one went busy with their lives. Coexistence or sharing ended with their resettlement. The occasions when everyone gets together in our home are very rare. A solid gathering became very limited, except for a day or two when they both came on leave once or twice a year. I miss togetherness, and I am not sure how long I have to wait again to feel the rejoice of a family get together.”*
(OA-13)

Although several participants missed residence, they never opposed their children moving abroad. However, with long years of separation and the deterioration of their physical abilities, their longing to be with their children has intensified. In response to the interview, an 81-year-old man stated that he would have asked his daughter to stay back if he had known that missing her would be so painful.


*“I accompanied my daughter to all the paperwork for migrating to Ireland and prepared all the documents. However, still, she is not here with me and is always a pain, no matter how much I hide. I never opposed her decision to migrate, but I did not realize that her absence would be felt deeply. If I had known living apart would be this hard, I might have asked her to stay back.”*
(OA-2)

### 3.3. Feeling Lonely, Neglected and Stagnant

Missing coresidence seems to make participants feel isolated, leading to a sense of loneliness and neglect in later years of solitude. The participants who aged in solitude experienced neglect and loneliness in their twilight years, which in turn made them anxious. In addition to distance from children, the absence of visitors, the loss of spouses and minimal support from relatives and neighborhoods often contribute to feelings of loneliness and neglect. In this context, one of the participants, who was partially bedridden and living alone in his 80s, stated the following:


*“There is no one to look after me. Food for me has been arranged at the next house. The boy will come and give food every day. …There is no one. I am alone. As the next house has been told to provide food, the food will be brought on time. There is no one to look forward to except the boy who brings the food. Then, my sister and brother-in-law will come up very rarely. Apart from that, no one else will come to see me. There is no one to make time for me. Why bother others when you know there is no one to care for me?”*
(OA-3)

For several participants, having no one to make even an occasional visit can feel extremely stressful. Although relatives live nearby, many participants reported feeling lonely due to a lack of support and having no visitors intensified their monotonous lifestyle. An 81-year-old female reported how she experiences loneliness and stagnancy when there is no one to meet and talk to.


*“I wish some relatives or friends could visit me often. However, no one comes. I hardly remember having visitors here in the recent past. I feel so lonely at times. Sitting still, doing the same things every day, feels stagnant.”*
(OA-5)

The feeling of loneliness is reported to be intensified by the grief of losing a spouse, especially for participants who have been passive in managing their home and finances. For a widowed participant, the loss of a spouse feels devastating both practically and emotionally.


*“The loss of a partner is incomparable, no matter who else is there to support you. With his death, it felt like I lost almost everything I had. He is the one who used to take care of the whole family and business matters. Neither I nor my children ever knew anything about these affairs. Now, suddenly, all these have become my responsibility. …With his departure, I’m now learning many things I never knew, and it is never easy to do everything all alone at this age.”*
(OA-18)

### 3.4. Feeling Anxious While Living Alone

The anxiety of being left behind seems to have increased with the deteriorating health conditions of oneself or one’s spouse. Participants who could not manage contingencies feel overwhelmed while living apart from their children. A similar experience was shared by one of the participants as follows:


*“Every time my husband falls sick, I wish my children were close by so that they do not have to rush overnight. I feel anxious until they come home and things get better. On a normal day when things are not worse, I am not worried much about the distance. If things go wrong, I am a little helpless to do things alone.”*
(OA-17)

Several participants feel anxious and insecure, anticipating the uncertainties that may arise at any time while living alone. Imagining the worst scenarios and the perceived care gap while living alone intensified the feeling of anxiety, as mentioned by one of the participants.


*“Having no one turn to in case something happens to us makes me anxious very often. I will try to calm myself by thinking that since our son is from Bangalore. He can reach here overnight. However, he is also trying to leave the country. … The most important challenge for me is the anxiety that comes with being alone. The feeling of being left alone creates a threat, and I always tend to imagine all the worst scenarios that can happen to us. I fully understand that they will have to step aside for work-related purposes. However, their absence is haunting me horribly.”*
(OA-15)

Although aging alone presents several emotional and practical challenges, most participants prefer not to burden migrant children with their fears and uncertainties that cannot be solved from a distance. Fear of being a burden to children makes them anxious. One of the female participants responded as follows.


*“What bothers me is not to trouble them when they are busy with their lives. It is our duty to move on with our lives when we already know they cannot do much from a distance.”*
(OA-17)

Similarly, another participant who has been a blood cancer patient for the last 17 years wishes not to trouble anyone around her because of her personal or health-related needs.


*“Even when I get older, I have the desire not to trouble anyone, especially my children abroad. I have been a blood cancer patient for the last 17 years, and I am still under medication…Watching someone suffer because of my needs is my worst fear.”*
(OA-8)

### 3.5. Difficulties in Accessing Emotional and Instrumental Care Needs

In addition to emotional deprivation, several practical inconveniences are experienced by left-behind older adults in long-distance care. Several participants experienced difficulties in managing hospitalizations, challenges in fulfilling day-to-day chores, inability to handle contingencies, and difficulty maintaining extended family ties while aging alone. Examples of similar experiences were stated by the participants as follows.

Managing daily life seems challenging for several participants, especially when their health deteriorates to some extent. A substantial number of participants reported limited mobility, an inability to travel alone, and a lack of familiarity with administrative work related to banks, insurance, and pensions, seemed to intensify the practical deficits while growing old alone. Several participants stated that these deprivations could be managed if their children had not migrated.


*“Now, today itself, I went alone for my mother’s pension. It was raining heavily. The office was on the third floor, and it was difficult to climb the steps, but there was no other way. If there were any children with me, it would be enough to give me an authorization letter to get things done. It is difficult to go out and climb the steps alone. However, no one else is there to do it on my behalf. No matter who is there around, certain emergencies are too difficult to handle alone.”*
(OA-13)

For most interviewed participants, hospitalizations seem to be a challenge, as there is no one to take them to hospitals for their routine check-ups or at times of hospital admissions, as stated by both male and female participants.


*“The most important crisis I have ever had is having to go to the hospital and visit. …Having no one to take you to the hospital is a real crisis. If my children were here, they would have taken me whenever I was due for my regular consultation. My biggest concern is about the hospitalizations that can pop up at any time.”*
(OA-4)

Apart from hospitalization and health contingencies, the emotional unavailability of children to address other family problems seems to create a care deficit among several participants, as stated by a participant.


*“Although they have been gone for many years, our way of life has changed significantly. Not having children around can be difficult, especially when there are family problems. The pressure of problems at home and not having them around to share is an enormous hardship.”*
(OA-13)

Although physical distance from children results in several deficits, having a physically capable and supportive partner seems to enhance the sense of comfort and security in migrant households. Examples of spousal support in managing day-to-day life and hospital emergencies were stated by several participants as follows.


*“My husband is very active and healthy enough to look after things. We support each other if there is anything like any hospitalizations or travel requirements. Hence, life feels manageable. Therefore, I have not felt the need for someone else to look after us.”*
(OA-17)


*“The support of my partner is very important… Even when children are apart, things feel manageable thus far because she does everything I want. We take care of each other, accompany when there is a hospital check-up, and travel together if there is anything outside the home.”*
(OA-13)

In addition to the distance from the child, knowing that even his wife would not be able to rush him to the hospital in case of an emergency makes him vulnerable. He stated as follows.


*“If I need something, neither my wife nor my brother can come to help me. Because both are already sick. If I had a heart attack at night, I would probably die at home. If I am not in a condition to make a phone call by myself, no one will take me to the hospital for sure. … I am aware of the kind of care deficit I might encounter with the deterioration of physical capabilities.”*
(OA-20)

Similarly, several participants experienced the inability to attend family functions, funerals, and other emergencies in extended families, which deteriorated their relationships and close-knit ties. In numerous situations, technology failed to replace human involvement, and the absence of children during critical moments heightened the participants’ anxiety and vulnerability.


*“What I found to be the greatest difficulty was the emergencies that came up when the children were away from home. Recently, a relative who is very close to us passed away. As it is a faraway place, we cannot go there alone, but if we do not go there, the relationship will break. It creates enormous emotional difficulty. If we have children, we can at least send them as our representatives. Or at least I can ask them to accompany me. Technology cannot replace people, even if there are so many advancements around.”*
(OA-13)

Although, older adults in migrant housheolds perceive care relationships through virtual connections and alternative provisions. They experience a deep sense of loss, loneliness, and induced anxiety while aging alone, which is further intensified through instrumental care needs that remain unmet in trans local and transnational families. The absence of intergenerational coexistence, limited support from extended families and neighborhoods, and the resulting barriers in responding to emergencies are reported to intensify the vulnerabilities of aging alone.

## 4. Discussion

This study aimed to explore the perceptions and experiences of left-behind older adults regarding solitary living and vulnerabilities due to remote care in migrant households in Kerala. The thematic analysis revealed their perceptions and experiences of aging alone outside intergenerational coresidence, illuminating five key themes: cues of care from a distance, missing family coexistence, feeling lonely and neglected, experiencing anxiety while living alone, and deficits in instrumental and health needs. Our analysis emphasizes the perceptions and experiences of older adults receiving long-distance care, illustrating the complex realities of vulnerability in old age, which arises from emotional and instrumental care deprivation that affects daily functioning and well-being. Despite virtual and occasional in-person support from caregivers, care deficits tend to overshadow these cues of care, resulting in increased vulnerabilities in long-distance care.

While existing evidence confirms a remarkable decline in the physical and emotional well-being of older adults who age alone [37], our findings stand out by offering insights into the perceived cues of care from a distance. Despite being geographically separated, several participants recognized their children’s efforts to care for them through virtual connections and in-person care at times of contingency, making distance bearable to an extent. Efforts from migrant children to maintain regular contact, arrange alternative care provisions, and check the health status of their older parents are perceived as signs of care even when their physical proximity is compromised. Previous findings on nurturing long-distance care include regular phone calls, letters, emails, and text messages as practical ways of exchanging care with families abroad [21,38]. However, our findings shed light on the duality of experience, where technologies have increased the intensity of virtual care [39,40], while they fall short in addressing the emotional void of intergenerational separation, highlighting the perceived failure of technologies in substituting for humans in care. Similarly, a significant association was observed between internet adoption and social inclusion [41], even when there is continuing ambiguity in providing effective virtual care [3]. Although our analysis underscores cues of care as pivotal in managing relationships from afar, [42] argue that minimal distance from the migrant child and absence of disability are essential to receiving quality informal care. However, our analysis emphasized that distance is manageable for participants who receive spousal support to compensate for the absence of their children. Thus, our analysis argues that having a functional spouse makes a difference in the narrations of care in their later years, suggesting the pivotal role played by spousal support while navigating the challenges of remote caring in later years. Conversely, the experience of remote care is often worsened by the grief of losing spouses, leading to heightened emotional instability, anxiety and difficulties in coping during later years of solitary living. Regardless of gender, spousal loss seemed to impact older adults; however, compared with older women who lost their spouses, older men were observed to experience long-term loneliness and emotional deficits. This could be due to the alternative close-knit family ties that women are more willing to access while they age alone. Our findings on spousal support are in line with the observations of Ugargol & Bailey, (2021) who argue that reciprocity between spouses is rooted in the obligation and responsibility built on the institution of marriage [7].

In line with the previous observations, our analysis of the experiences of long-distance care confirms the feeling of missing intergenerational coexistence where older adults share warmth, love and care across generations. Missing children and family togetherness intensify loneliness and feelings of neglect in later years of solitude, especially when contacts are reduced and alternative engagements are limited while individuals age alone [43]. Our analysis deepens the narrative of experiencing distance and consequential care deficits, long distances from children, limited support from extended family members, and a lack of visitors to spend time with them. Similar conclusions were drawn by Thapa et al., (2018), suggesting that the absence of children can trigger stress-related ailments, leading to emotional ambivalence, distress and anger in twilight years, presenting a multifaceted vulnerability and need for mental health nursing services while aging apart [14]. Existing conceptual assumptions on loneliness in later years suggest that the physical presence of kin, emotional closeness with children afar, and expectations about fun, generativity and the gap between these hopes and reality are pivotal in shaping experiences of loneliness, suggesting a greater risk of loneliness and social vulnerability among left-behind older adults [24]. Thus, echoing prior studies, our findings present a framework of care deficits where older adults who age alone miss the normative pattern of coexistence, leading to feelings of loneliness, which in turn contributes to higher odds of anxiety and vice versa. Similarly, the perceived difficulties in fulfilling functional needs often exacerbate emotional deficits, indicating mutual causality of emotional and instrumental care deprivations. Comparable findings have been reported by Dakua et al., (2023), implying a risk of mixed loneliness and depression, fear of being alone, and apprehension about aging as the core concerns of growing old outside intergenerational coresidence [44]. Notably, our study presents the nuanced intersection of filial piety and autonomy in later years, indicating the need for complex care negotiations in migrant households. While participants expect reciprocation from their kin through virtual cues, they fear being a burden to their children and prefer not to share their concerns about living alone, which in turn makes them anxious. The findings partially mirror the conceptualizations of Ugargol & Bailey, (2021) on filial piety, while also introducing the preference of older adults to be reticent and experience autonomous aging [7]. This divergence could be due to differences in personal preferences for aging, increased financial remittance, and alternative support available in the homeland, suggesting evolving conceptualizations of care obligations.

Another major theme that evolved during the analysis was care deprivation caused by the absence of their kin, coupled with declining physical functionalities. Migration leads to reduced family time, the unavailability of children at times of emotional and personal difficulties, and the absence of a family representative to fulfill the needs of the homeland. Older adults in migrant housheolds often struggle in getting things done, which could have been easily fulfilled in intergenerational coresidence, leading to feelings of anxiety and incompetence. Our findings are in line with the arguments that distance from children induces functional deprivations among left-behind older adults and that they struggle to meet their day-to-day needs [45,46]. Notably, our analysis revealed that most of the participants viewed hospitalizations and medical emergencies as major casualties in remote care, indicating weak support structures to respond at times of casualty, to assist during hospitalizations, and to provide home care after hospital discharge. However, Braimah et al., (2024) argue that the experience of aging is subjective and unique, depending on the availability of financial remittances and other support structures, where the convergence of individual and broader social systems is vital, irrespective of the living arrangement, suggesting the need for a support system within and outside the family [38,43]. Interactions with the community and neighborhood reduce the risk of isolation and neglect while providing avenues to foster socialization and positive mental health in later years [46]. While previous evidence argues that social networks among older adults help in managing both planned and unplanned hospitalization during the twilight years [47], our findings, conversely, reveal that left-behind older adults often experience a loss of support and care from their microsystem and that they long for family togetherness, implying the need to strengthen meso-systems to support older adults living alone. While existing evidence argues that educational status, functional capabilities, the presence of comorbidities, and gender have been found to have a profound impact on shaping the experiences of vulnerability while aging alone [12,29,48], corroborating existing evidence, our analysis presents a complex interplay of functional and emotional needs in shaping the perceived vulnerabilities of aging alone. Reduced mobility, a lack of accompaniment from children, and the inability of children to act on their behalf during familial crises are a few inevitable occasions that transcend beyond the scope of virtual care, where technologies fail to address the care gap posed by distance. However, the reciprocity of care through virtual means has helped older adults feel wanted, loved, connected, and supported in later years of solitude, even though they have fallen short in substituting the essence of filial piety through in-person care.

While care cues are regarded as pivotal in dealing with distance from their migrant children, several older adults cope with solitude through proactive strategies such as pursuing hobbies, practicing emotional regulation, strengthening social networks, building a strong sense of neighborhood and maintaining intergenerational connections through various virtual platforms [14,49] Existing evidence further confirms that strong social networks often increase cognitive and physical functional abilities by reducing the mortality risk associated with aging alone [18]. While care obligations are profoundly strong in the Eastern context, further research is warranted to investigate the correlation between cultural expectations of care and its impact on caregivers [50], opening further discourses into mental opportunities and challenges in mental health nursing services for older adults who age alone. Thus, our findings contribute to the culturally sensitive narrative on care, revealing that even from afar, care and vulnerability are deeply negotiated, emotional, and ever-evolving, and are shaped by the directional causality of emotional and practical care needs, suggesting innovative approaches to include mental health care services in the broader public health care system for community-dwelling older adults.

## 5. Strengths and Limitations of the Study

Employing an inductive approach, our study sought to explore the perceptions and experiences of functional deprivation among left-behind older adults in India. Three distinct geographic regions in Kerala, Pathanamthitta, Thrissur, and Kannur, were purposively selected to account for the regional variations in aging across southern, middle, and northern Kerala, respectively. By integrating the rubout theoretical underpinning, our study accounts for the dynamic notion of complex care realities and functional deprivations in the transnational and trans local households of Kerala. While the findings offer valuable insights into the experiences and perceptions of functional deprivation and care conceptualizations, the scope of the study was limited to older adults in migrant households, leaving ample room for further investigation into the experiences of older adults residing with extended family members and those in institutional care arrangements. Furthermore, our study could not capture the perceptions of older adults from the Muslim community, as most of the families co-resided with either daughters-in-law or relatives despite the migration of their adult children, making it difficult to identify families falling within the inclusion criteria. Hence, further research could explore the narrations of functional and emotional deprivations across various care arrangements and communities to capture the evolving perspectives on care that cross borders. Future investigations could explore the intersection of self-rated health and perceived vulnerability in aging alone to understand the nuances of care realities among older adults across different functional abilities.

## 6. Conclusions

The intersection of the migration and nuclearization of families poses complexities in familial dynamics in trans local and transnational families, indicating a greater risk of emotional, functional, and social deprivation among left-behind older parents. Emotional deprivation and vulnerability due to aging alone often challenge health-promoting behaviors and population-level determinants of aging well. Despite several advancements, the distance from migrant children continues to make older adults feel anxious, miss their family togetherness, and experience increased loneliness, neglect, and care gaps in later years, contributing to a multifaceted causality of vulnerabilities while aging alone. Narratives of distance care are often shaped by the bidirectional flow of care across generations through virtual and in-person means, where emotional and functional deprivations continue to challenge the quality of informal distant care among left-behind older adults. The insights into care cues warrant the adoption of social and health care policies that integrate person-centered care plans for older adults who age alone. Furthermore, clinical practices should promote strength-based interventions that leverage the existing coping mechanisms and resilience of older adults. Efforts to safeguard mental health among community-dwelling older adults are crucial for sustaining their functional capacity, thereby delaying psychological morbidities and cognitive decline during aging. Future research should investigate positive coping among left-behind older adults and design community-based mental health programs within public health and social care frameworks that align with the objectives of healthy aging and the Sustainable Development Goals.

## Figures and Tables

**Figure 1 ijerph-22-01745-f001:**
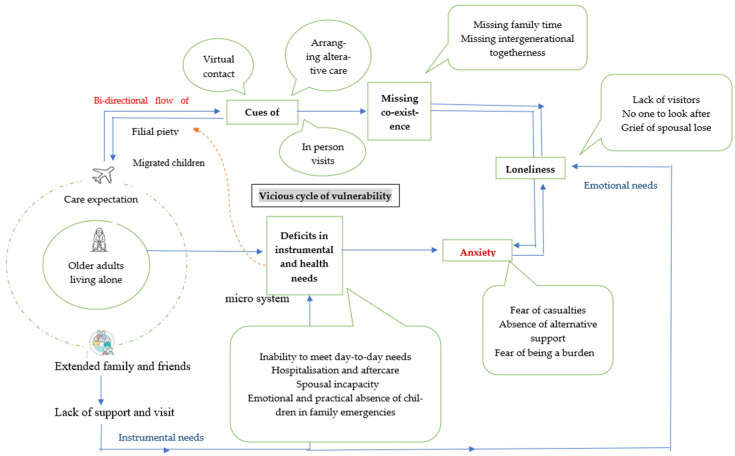
Pathways to emotional and functional care deprivation among older adults in migrant households.

**Table 1 ijerph-22-01745-t001:** Sociodemographic characteristics of the participants.

Participant	Age Group (in Years)	Religion	Employment Status	Marital Status	Medical Conditions	Health Status	Living Arrangement	Number of Children	Type of Migration	Availability of Paid Caregiver
OA-1	70–75	Christian	Returned from Abroad (Private sector)	Widower	Yes	Healthy and mobile	Living alone	2	Both internal and international	No
OA-2	80–85	Christian	Retired private sector employee	Widower	No	Healthy and partially mobile	Living alone	2	International	Yes (full-time caretaker)
OA-3	80–85	Christian	Retired Public sector employee	Widower	Yes	Partially mobility	Living alone	4	Internal	No
OA-4	75–79	Christian	Homemaker	Widow	Yes	Healthy and Mobile	Living alone	1	International	Yes (full-time caretaker)
OA-5	80–85	Christian	Retired private sector employee	Widow	No	Healthy and partially mobile	Living alone	2	Both	Yes (full-time caretaker)
OA-6	80–85	Christian	Returned from abroad (Private)	Widow	Yes	Healthy and Mobile	Living alone	2	Internal	No
OA-7	65–69	Hindu	Retired Private sector employee	Married	No	Healthy and mobile	Living with spouse	2	Both	No
OA-8	65–69	Hindu	Homemaker	Married	No	Healthy and mobile	Living with spouse	2	Both	No
OA-9	75–79	Christian	Retired Public sector employee	Married	Yes	Healthy and mobile	Living with spouse	2	both	No
OA-10	70–75	Hindu	Retired private sector employee	Married	Yes	Partially mobile	Living with spouse	2	Both	No
OA-11	70–75	Hindu	Retired Public sector employee	Married	Yes	Healthy and Mobile	Living with spouse	2	Both	No
OA-12	80–85	Hindu	Retired private sector employee	Widower	Yes	Partially mobile	Living alone	3	Internal	Yes (Part-time)
OA-13	65–69	Christian	Retired Public sector employee	Married	No	Healthy and mobile	Living with spouse	2	Internal	No
OA-14	65–69	Hindu	Retired Public sector employee	Married	No	Healthy and Mobile	Living with spouse	2	Both	No
OA-15	65–69	Hindu	Independent practitioner	Married	No	Healthy and Mobile	Living with spouse	2	Both	No
OA-16	70–75	Hindu	Retired public sector employee	Married	No	Healthy and Mobile	Living with spouse	3	internal	No
OA-17	65–69	Hindu	Home maker	Married	No	Healthy and Mobile	Living with spouse	2	Internal	No
OA-18	65–69	Christian	Homemaker	Widower	Yes	Healthy and Mobile	Living alone	2	International	No
OA-19	70–75	Hindu	Retired public sector employee	Married	No	healthy and Mobile	Living alone	2	Both	No
OA-20	70–75	Christian	Returned from Abroad (Private sector)	Married	No	Healthy and mobile	Living with a spouse	2	Both	No

**Table 2 ijerph-22-01745-t002:** Summary of themes and codes related to care experiences among left-behind older adults.

Theme	Codes
Cues of care from afar	Maintaining virtual contact, alternative care provisions arranged by children, in person, during contingencies
Missing coexistence with multi-generational families	Missing the physical presence of children and grandchildren, losing intergenerational togetherness,
Feeling lonely, neglected, and stagnant	Lack of visitors, no one to look after, enduring spousal loss, and its aftermath
Feeling anxious while living alone	Anxiety about managing contingencies, absence of alternative support to rush at need, fear of being a burden
Difficulties in accessing emotional and instrumental care needs	Challenges in fulfilling day-to-day chores, inability to handle contingencies, and hassles in maintaining extended family ties, deriving mutual support in later years

## Data Availability

The datasets generated and analyzed during the current study are available from the corresponding author upon reasonable request.

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
