# Peer review of "‘Care Beyond Co-Residence’: A Qualitative Exploration of Emotional and Instrumental Care Gaps Among Older Adults in Migrant Households of Kerala"

_ijerph, 2025, doi:10.3390/ijerph22111745_

Round 1
Reviewer 1 Report
Comments and Suggestions for Authors
The article “Lived Realities of Emotional and Instrumental Care Gaps Among Left Behind Older Adults in Kerala” presents a timely and relevant qualitative study on the lived experiences of older adults in Kerala whose adult children have migrated. The research explores a very important contemporary demographic phenomenon: the interconnection between population aging and youth migration.
The article represents a valuable contribution to its scientific field. Its main strength lies in its qualitative narrative approach, which deeply captures the perceptions of the elderly, offering an understanding that quantitative studies often fail to achieve. The focus on the specific context of Kerala, a state with high emigration rates and a growing elderly population, makes it an interesting case study. Furthermore, the study presents a balanced analysis, not only focusing on gaps but also exploring the "Cues of Care from Afar," thus acknowledging the complexity of transnational family dynamics.
The article's structure is clear, logical, and well-interconnected. The introduction effectively contextualizes the problem, the literature review is well-conducted with recent authors, the methodology is described transparently, and the results are well-presented, supported by direct quotes from the participants. As noted, the excellent cohesion of the text is a significant strength.
Although the article appears to be well-structured and organized, we suggest some improvements, which we will now detail:
- Figure 1, which illustrates the pathways to care deprivations, is a strong point of the analysis. However, the use of the term "vicious cycle of vulnerability" to describe these dynamics may be an oversimplification. Qualitative data, while excellent for demonstrating the interconnection between factors like anxiety and instrumental deficits, can rarely prove rigorous cyclical causality. We suggest using more precise terminology that better reflects the associative nature of the findings.
- The article uses the expression "'loneliness pandemic.'" While effective in conveying the seriousness of the problem, this term may be perceived as more journalistic than academic. Its replacement with a more neutral alternative is recommended, given that this is a scientific article intended for publication in a high-impact journal.
- The authors transparently acknowledge that the study could not capture the perceptions of the Muslim community. This issue led us to consider the importance of including a brief reflection on the purposive sampling recruitment process. Essentially, describing how participants were contacted (e.g., through community leaders, the snowball effect, etc.) would help readers assess potential selection biases and would increase the study's methodological transparency.
- The article's focus is on "gaps" and "deficits." We therefore propose a small addition to the Discussion section that addresses the resilience and coping strategies that the elderly develop, beyond valuing the "cues of care.” This would provide a more holistic view of the lived experience and point to promising avenues for future research focused on positive interventions.
In summary, in our view, this is a very good article. Nevertheless, the suggestions presented here are intended to refine the work.
Given the above, we recommend the acceptance of this article, conditional on the implementation of the suggested minor revisions.
Reviewer 2 Report
Comments and Suggestions for Authors
Studying the quality of life of older adults who live alone due to their children's immigration is valuable, and this study was conducted systematically with that goal in mind. With further refinement in the analysis and writing, I believe it will be readily accepted for publication in an academic journal. Here are somethings I would like you to improve it:
- Please revise the title to indicate that this study is qualitative.
- Since the characteristics of the study participants are crucial for understanding the results, please place the table in the main text of the manuscript, not in the appendix, and provide a more detailed description of the participant characteristics.
- Please provide more specific details on your qualitative research method and explain the theoretical basis for your analysis and interpretation, i.e., the qualitative research method you used.
- Based on this qualitative analysis, you presented Pathways to Emotional and Functional Care Deprivations in Figure 1, but the causal relationship appears artificial. In other words, presenting such a diagram, while the evidence is clear, seems inappropriate. If this is to be presented as is, further evidence is needed.
- Please provide more specific implications of these research findings for health, social care, or clinical practice.
